# Formation and Maintenance of Tissue Resident Memory CD8+ T Cells after Viral Infection

**DOI:** 10.3390/pathogens8040196

**Published:** 2019-10-18

**Authors:** David J. Topham, Emma C. Reilly, Kris Lambert Emo, Mike Sportiello

**Affiliations:** David H. Smith Center for Vaccine Biology and Immunology, Department of Microbiology and Immunology, University of Rochester Medical Center, Rochester, NY 14642, USA; emma_reilly@urmc.rochester.edu (E.C.R.); kris_lambert@urmc.rochester.edu (K.L.E.); michael_sportiello@urmc.rochester.edu (M.S.)

**Keywords:** T cell, memory, tissue resident, immune response

## Abstract

Tissue resident memory (T_RM_) CD8 T cells comprise a memory population that forms in peripheral, non-lymphoid tissues after an infection that does not recirculate into the bloodstream or other tissues. T_RM_ cells often recognize conserved peptide epitopes shared among different strains of a pathogen and so offer a protective role upon secondary encounter with the same or related pathogens. Several recent studies have begun to shed light on the intrinsic and extrinsic factors regulating T_RM_. In addition, work is being done to understand how canonical “markers” of T_RM_ actually affect the function of these cells. Many of these markers regulate the generation or persistence of these T_RM_ cells, an important point of study due to the differences in persistence of T_RM_ between tissues, which may impact future vaccine development to cater towards these important differences. In this review, we will discuss recent advances in T_RM_ biology that may lead to strategies designed to promote this important protective immune subset.

## 1. Introduction

As early as 1994, Walter Gerhard’s lab demonstrated that protection from a serologically distinct strain of influenza in immune mice was largely due to cross-reactive CD4 and CD8 T cells, with CD8 T cells having the greatest effect in the airways and lung [1]. This form of immunity, called heterosubtypic protection, was effective but short-lived and waned over several months. In 2001, David Woodland’s lab used a different respiratory pathogen, Sendai (parainfluenza) virus, to demonstrate that functional virus-specific CD8 T cells persisted in the airways and lungs. However, they observed a similar decline in numbers over a year after infection [2], and the number of virus-specific CD8 T cells in the airways correlated with protection. These studies pre-date the identification of cell surface markers used to define T_RM_ cells, although Hogan et al. did measure CD69 expression [2]. In the same year, David Masopust and Leo Lefrancois used major histocompatibility complex (MHC) tetramers carrying viral or bacterial-specific peptides to show that after a systemic infection, pathogen-specific memory CD8 T cells localized to all peripheral non-lymphoid after infection [3]. In 2004, inspired by this data, our lab used an influenza virus model to demonstrate that Very Late Antigen-1 (VLA-1) (the CD49a/CD29 α1β1 integrin heterodimer) was essential to maintain virus-specific CD8 memory T cells in the airways and lung tissues [4]. In that same study, we found virus-specific CD8 T cells expressing VLA-1 in every peripheral tissue examined, showing that infection of a tissue is not a prerequisite for T_RM_ localization. Overall, these results indicate that infection results in populations of CD49a+, CD69+, CD8+ memory T cells that reside in peripheral tissues: tissue resident memory T cells. [4]. Given the importance of this memory T cell population to secondary immune protection, here we review recent advances in the understanding of T_RM_ and call for further research that will lead to strategies designed to improve local tissue immunity.

## 2. Markers Used to Define T_RM_


Expression of several cell surface proteins have been found to be common among T_RM_ from different tissues. Absence of chemokine receptor 7 (CCR7) and expression of CD69 were identified early [5,6], and CD49a or VLA-1 came next [4]. At the time, reagents to study mouse CD49a were limited in availability, so this marker was not initially widely adopted. In 2005, CD103 expression was found to define a population of memory T cells in the tonsil [7] and in 2010, the Bevan lab associated CD103 with T_RM_ in the brain [8]. Since then, CD69 and CD103 expression have been the most widely used markers of T_RM_, although more recent studies in both mice and humans have demonstrated the importance of CD49a. It may be necessary to use these three markers (CD69, CD49a, CD103) in combination when studying T_RM_ from different tissues, as it is unlikely to be a homogenous population as previously assumed. Many studies of function and gene expression of T_RM_ have relied on either CD69 or CD103 alone or in combination [9,10]. Although CD69 is critical for T_RM_ in some peripheral tissues (e.g., kidney), it does not appear to be a universal requirement for T_RM_ in all sites [11]. Our lab has recently discovered that using CD49a and CD103 in combination identifies up to four potentially distinct memory T cell subsets in the lungs and airways (including trachea), although cells expressing only CD69 and CD103 do not appear to persist long-term [12]. Using all three markers may be necessary to paint the full picture of cell diversity, function, and gene expression profiles (Table 1).

## 3. Functions of Memory T Cell Markers

The cell surface proteins used to define T_RM_ and other memory subsets are not just markers; they have functions, though relatively little has been done to discern these functions [13]. Naïve T cells lack expression of CD49a and CD69 and have only low levels of CD103 on their surface. Instead, they can be defined by high expression of CD62L (L-selectin), low expression of CD44, and expression of CCR7 [14]. CD62L is a ligand for a receptor expressed on high endothelial venules to help direct naïve T cells to secondary lymphoid organs (SLOs) [15]. CCR7 serves a similar purpose by receiving CCL19 and CCL21 chemokine signals coming from SLOs [16,17]. CD44 binds hyaluronic acids in the extracellular matrix of almost all tissues and may itself be important for regulation of T cell motility, positioning, and retention in peripheral and lymphoid tissues [18,19].

“Central memory T cells” (T_CM_) are CD44^high^, CD62L^high^, and retain CCR7 expression [14] and lymphoid homing potential. “Effector memory T cells” (T_EM_) are CD44^high^, CD62L^low^, and CCR7^–^, and preferentially circulate though the bloodstream and into through peripheral non-lymphoid tissues [14]. T_EM_ form one of the four memory CD8 T cell subsets found in lung and airways, and do not express CD49a, CD103, and CD69. CD49a/VLA-1 is the only known integrin ligand for collagen IV, uniquely located in the lamina densa of barrier tissues, though it has been shown to also bind collagen I [20]. It provides anti-apoptotic signals to the T cells when bound to collagen [21] and has proven essential for maintaining T_RM_ in peripheral tissues. T cells that express CD49a/VLA-1 tend to localize close to or within the epithelial cell layer where collagen IV is abundant [22]. CD69 is an S1P1 antagonist, limiting signals that direct lymphocytes into draining lymphatics [23,24]. It is also a ligand for Galectin-1 (Gal-1) and may have a role in cell–cell and cell–matrix interactions [25]. CD103 is a ligand for E-cadherin, which is most widely expressed in the junctions between epithelial cells at mucosal and barrier sites [26]. Both CD8 T cells and dendritic cells (DC) can also express this ligand for CD103, two more sources of potential cell–cell interaction [27]. CD103 is also expressed on a subset of DC in the airways that serve to sample the epithelium for antigen by interaction with cell junctions [28], possibly playing a similar role for CD8 T_RM_ cells in epithelial surveillance. Expression of these adhesion and migratory receptors may position memory T cell subsets in different locations in the mucosa depending on ligand availability (Figure 1, Table 2)

## 4. Formation of CD8 T_RM_ and Interactions with Other Cells in the Tissue

Virus-specific CD8 cells that express CD49a do not appear in the tissue until after the infection is cleared [4]. In influenza infection, CD8 T cells in the tissue express CD49a, CD69, and CD103 by day 14, suggesting cells with a T_RM_ phenotype develop relatively early as the tissue recovers from infection [12]. There has been much speculation regarding when and where T_RM_ cells form: Takamura demonstrated that in areas of repair after influenza infection, there are repair-associated memory depots (RAMD) containing populations of keratin-5+ cells that express alpha-V integrins [29]. Alpha-V integrins can activate latent TGF−β in the tissue, a necessary cytokine for T_RM_ to express CD49a and CD103 [30,31]. This suggests the T_RM_ cells form directly in the tissues they take residence in.

However, this hypothesis does not explain the early observations that pathogen-specific T_RM_ can be found in all peripheral tissues examined regardless of whether they were directly infected [4]. Experiments using FTY720 to block S1P signaling and inhibit lymph node (LN) egress resulted in an accumulation of CD8 T cells bearing the T_RM_ phenotype in the lymph nodes (Topham unpublished). In such an experiment it is difficult to discern in situ differentiation of T_RM_ in the LN from drainage of the cells out of the tissues, but it does question whether T_RM_ cells solely develop in the recovering tissue, and also suggests there may be a window after infection during which T_RM_ cells disperse systemically to peripheral tissues. Takamura’s parabiosis experiments suggest that mice need to be paired by day 6 after infection to show distribution of virus specific cells to lungs and other tissues of both mice [29]. However, more focused tracking experiments will need to be done to concretely answer this question.

Additional signals in the tissue may be needed to generate T_RM_. Using a push-pull strategy, the Kohlmeier lab recently demonstrated that encounter with antigen in the tissue may be required for T_RM_ localization [32]. Using an intramuscular infection model and intranasal administration of CpG or CpG plus nucleoprotein (NP) peptide, only the animals given local peptide developed substantial NP-specific T_RM_ cells in the respiratory tract [32]. Antigen encounter has been shown to down-regulate CCR7 on recently activated T cells perhaps explaining the need for antigen to be present for optimal T_RM_ formation [33,34,35].

A role for IL-10 has also been indicated in the production of monocyte-derived TGF−β; a critical mediator for upregulation of CD103 (and possibly CD49a) in several studies [36,37,38]. These experiments used various adjuvants and antigen in vivo, and an in vitro model of T cell activation to induce an immune response. With this system, T cells exposed to IL10 and TGF−β very early in priming had the highest levels of CD103, suggesting T_RM_ precursors could be generated in the LN [38]. This scenario of early exposure to these cytokines is less likely to occur in models of infection given the slow kinetics (days) and levels of IL-10, which are not high early in a flu infection [39], but antigen-bearing monocytes releasing Transforming Growth Factor (TGF)−β directly in relation to T_RM_ development has not been studied in the context of infection.

It is also possible that CD103 could be expressed prior to other T_RM_ markers, and inhibition of CD103 leads to reduced numbers of T_RM_, consistent with a role in the accumulation in tissues [23]. Although these papers did not look at the function of CD49a, in a gut model of T_RM_, TGF−β was shown to be important for expression of CD49 on a4b7+ gut T_RM_ [37] In an influenza infection model, another stimulatory signal for T_RM_ formation comes from 4–1BB [40]. 4–1BB is a TNF family receptor and endogenous signals from antigen-bearing 4–1BB ligand expressing cells in the tissue enhance T_RM_ formation while exogenous administration of 4–1BB ligand further expands the number of T_RM_ formed [40].

## 5. Metabolic Changes Associated with T_RM_

Tissues like skin, gut, and respiratory mucosa are very different environments than Secondary Lymphoid Organ (SLO)s and blood. The availability of nutrients such as glucose and oxygen vary compared to SLOs. The metabolic pathways utilized by T cells has been shown to vary with their state of activation [41]. As with most eukaryotic cells, naïve T cells primarily derive energy from oxidative phosphorylation in the mitochondria [42] to make adenosine triphosphate (ATP). T cell activation drives T cell metabolism toward aerobic glycolysis, the pentose phosphate pathway, and glutaminolysis [43]. These changes are associated with reprogramming of the metabolic transcriptome [44,45]. The different circulating memory T cells subsets utilize distinct metabolic programs. For example, central memory T cells primarily use oxidative phosphorylation, much like naïve T cells [42,46]. Effector memory T cells that can circulate through peripheral tissues use a more balanced combination of oxidative phosphorylation and glycolysis and have greater mitochondrial mass that makes them more comparable to effector T cells [47].

Peripheral tissues, especially barrier tissues and microcompartments contained within these offer different glucose levels, pH, and structural components, all of which can affect T cell metabolism and survival. T_RM_ cells are very active in the “resting” state as they constantly perform surveillance of the cells surrounding them for evidence of pathogen invasion. T_RM_ cells intimately interact with the components of their environment including epithelial cells and the extracellular matrix. T_RM_ cells abundantly express lipid receptors on their cell surface to provide them capacity for lipid uptake and metabolism [48]. These include fatty acid binding proteins FABP4, FABP5 to enhance lipolysis, low density lipid receptor, ApoE, and the CD36 scavenger receptor. These receptors facilitate uptake of free fatty acids (FFA) from the environment and these FFA can be used in oxidative phosphorylation and in fatty acid oxidation (FAO) [49]. T_RM_ can accumulate FFA in droplets within the cytoplasm just like T_EM_ are able to do [50], and these droplets can be coupled to distinct mitochondria upon activation [51,52]. T_RM_ may rely on traditional oxidative phosphorylation and glycolysis under “resting” conditions, and activate FAO by accessing the stored droplets upon encounter with pathogens and antigen recognition, however, this has yet to be directly tested.

## 6. Mechanisms that Suppress T_RM_ Activation May Have other Functions Important to T_RM_ Establishment and Maintenance

CD8 T_RM_ cells are armed and ready to protect against secondary infections. They can express high levels of Granzyme B giving them cytotoxic potential, secrete IFN−γ and TNF−α when activated, as well as a laundry list of chemokines to jumpstart immune cell recruitment [41]. Therefore, accidental or bystander activation of T_RM_ could result in serious, unwanted tissue damage. There are a number of inhibitory receptors expressed by CD8 T_RM_ that likely serve to keep these cells at bay until bona fide antigen recognition occurs through the TCR. Brain T_RM_ cells express PD1, a receptor associated with T cell exhaustion and inability to activate [53,54]. Blockade of PD-1 ligand in tumor settings greatly enhances T cell elimination of cancer cells [55]. How this suppression is overcome in T_RM_ during an infection is not known and may require a combination of “activating” signals. P2RX7 is a purinergic receptor expressed by T_RM_ [56]. Low ATP levels and signals through P2RX7 can promote T_RM_ survival by stimulating activation and metabolism. High signals, on the other hand, can drive T cell apoptosis [56], making the balance of signals in the environment of T_RM_ critical. T_RM_ cells also express CD244, a receptor previously described on Natural Killer (NK) cells [57]. Signals through CD244 can be either inhibitory or activating depending on intracellular levels of the adapter protein Signaling Lymphocytic Activation Molecule (SLAM) associated protein (SAP) [58]. SAP levels in T_RM_ are not known, but this knowledge could further indicate how these, and other receptors regulate T_RM_ maintenance and function. Collectively, these inhibitory molecules may keep T_RM_ in a state of readiness to respond, but inhibiting aberrant activation of T_RM_ by non-specific environmental cues.

## 7. Concluding Remarks

T_RM_ cells represent a very specialized subset of memory T cells that can be highly protective during secondary encounters with a previously seen pathogen. While not discussed here, they also seem to be important in controlling tumor cells and CD103 expression has been suggested as a predictor of tumor prognosis [59]. They have specialized functions that permit them to perform surveillance and protection at a number of barrier sites, including but not limited to skin, gut, respiratory tract, and female reproductive tract. However, many questions remain in our understanding of this unique cell subset. For example, we don’t fully appreciate why they seem to wane with time in some tissues, while they are continuously renewed in others. We also do not have effective, clinically feasible strategies of generating, maintaining, or improving their numbers. Live attenuated vaccines do generate a population of T_RM_ cells, however at a weakened capacity compared with natural infection [60,61]. Additionally, to date, T_RM_ cells are considered as one uniform population of cells and we believe that research examining T_RM_ subsets is lacking. For example, do all T_RM_ cells express CD69, CD49a, or CD103? Do populations exist that express different combinations of these markers? What are the genetic and epigenetic programs that drive their phenotype(s)? What are the mechanisms of controlling motility, persistence, and micro-localization within different tissues? In what ways, if at all, are T_RM_ cells specific for viruses different than those specific for bacteria, fungi, cancer, or other pathologies? These are all important questions the field needs to address to construct a more holistic model through which we cannot only understand memory and immunity, but use them to develop more effective vaccines and therapies.

## Figures and Tables

**Figure 1 pathogens-08-00196-f001:**
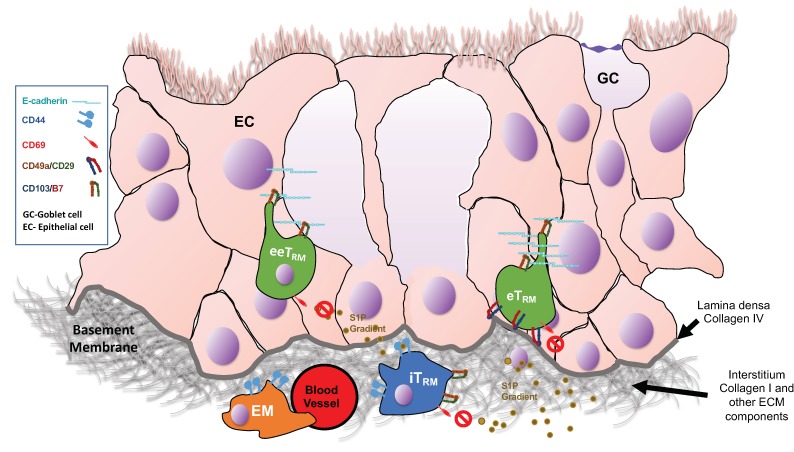
**Proposed tissue localization of CD8+ memory T cell subsets within epithelial tissue**. Based on the expression of adhesion molecules for ECM (CD49a, CD44) and, epithelial cells (CD103), and the location of their ligands, the memory cell subsets may be positioned in different tissue microenvironments.

**Table 1 pathogens-08-00196-t001:** Cell Surface Expression of Markers on Memory CD8 T cell subsets in the airways and lung tissue.

Memory type	Abbreviation	CD3	TCRb	CD8	CD44	CD62L	CD69	CD49a	CD103
Effector	EM	+	+	+	high	low	−	−	−
Epithelial T_RM_	eT_RM_	+	+	+	high	low	+	+	+
Interstitial T_RM_	iT_RM_	+	+	+	high	low	+	+	−
Epithelial Effector	eeT_RM_	+	+	+	high	low	+	−	+

Note: “+” indicates positive cell surface expression; “−” indicates the cell population is negative for cell surface expression “high” and “low” refer to the levels of cell surface expression.

**Table 2 pathogens-08-00196-t002:** Select markers of memory CD8 T cell subsets and their functions.

Marker	Function	Presence on T_RM_
CD44	Binds hyaluronic acids. May serve roles in motility and retention in both peripheral and lymphoid tissues.	+/High
CD62L	Ligand for high endothelial venules on SLO.	−/Low
CCR7	Chemokine receptor for S1P1.	−/Low
CD49a	Mediates adhesion to type I and IV collagen.	Positive on most T_RM_
CD69	Early activation marker. S1P1R antagonist. Binds Gal-1.	Positive on most T_RM_
CD103	Binds E-Cadherin.	Very often present, though can be tissue-dependent.

Note: “+” indicates positive cell surface expression; “−” indicated negative cell surface expression.

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
