# Peer review of "Formation and Maintenance of Tissue Resident Memory CD8+ T Cells after Viral Infection"

_pathogens, 2019, doi:10.3390/pathogens8040196_

Round 1
Reviewer 1 Report
The manuscript is well written and interesting.
In my opinion, 1. Introduction must contain a clearly described the aim of work (described in Abstract).
Author Response
Reviewer 1
The manuscript is well written and interesting.
In my opinion, 1. Introduction must contain a clearly described the aim of work (described in Abstract).
A statement of the aim of the review has been added to the introduction.

Reviewer 2 Report
Comments for the authors of the Pathogens manuscript number 614384: The authors of the Pathogens manuscript “Formation and Maintenance of Tissue Resident Memory CD8+ T Cells After Influenza Infection”, provide a timely review of the current understanding of the Tissue Resident Memory T cells. The manuscript provides information related to the history of this cell population, properties of the cells, and markers that are used to identify this cell population. Further, additional information explores the potential role of cellular receptors, in addition to their role as markers, and how these cells interact with host cells in tissue environments. While this review describes up-to-date findings for a relevant cells type, I have identified some items I would like the authors to address as they work toward improving the presentation of this information. General Comments: 1) The manuscript focused heavily on the T cell population, with less emphasis on the influenza virus contributions to the function of this cell population. The review seemed to focus on the cell population more, with a minor description of how influenza virus infection models were used to help describe some of the cellular functions. The suggestion would be to remove “…After Influenza Infection” from the title, unless the authors want to make a stronger case that the cellular phenotypes described are specific to influenza virus. It seems that the description of these cellular functions might be applicable for viruses other than influenza as well. 2) On page 2, line 72, the manuscript references a Table 1 that I was unable to find in the manuscript itself. 3) The manuscript would benefit from at least 1 Figure showing the markers and cellular functions associated with the markers. The same could be said for the cellular interactions described within the text. It was difficult to visualize the different cell types, roles in immunity, and cellular interactions using only the text provided. Specific comments 1) The manuscript would benefit from additional editing of the text, with emphasis on sentence structure. 2) On line 84, the word “CD62Llow” should be corrected to “CD62low”Author Response
Reviewer 2
Comments for the authors of the Pathogens manuscript number 614384: The authors of the Pathogens manuscript “Formation and Maintenance of Tissue Resident Memory CD8+ T Cells After Influenza Infection”, provide a timely review of the current understanding of the Tissue Resident Memory T cells. The manuscript provides information related to the history of this cell population, properties of the cells, and markers that are used to identify this cell population. Further, additional information explores the potential role of cellular receptors, in addition to their role as markers, and how these cells interact with host cells in tissue environments. While this review describes up-to-date findings for a relevant cells type, I have identified some items I would like the authors to address as they work toward improving the presentation of this information.
General Comments:
1) The manuscript focused heavily on the T cell population, with less emphasis on the influenza virus contributions to the function of this cell population. The review seemed to focus on the cell population more, with a minor description of how influenza virus infection models were used to help describe some of the cellular functions. The suggestion would be to remove “…After Influenza Infection” from the title, unless the authors want to make a stronger case that the cellular phenotypes described are specific to influenza virus. It seems that the description of these cellular functions might be applicable for viruses other than influenza as well.
We have changed the title from “Influenza” to “Virus” to make it more generalizable
2) On page 2, line 72, the manuscript references a Table 1 that I was unable to find in the manuscript itself.
Somehow, while the table was included in the manuscript submission, it did not make it to the proofs. This will be corrected.
3) The manuscript would benefit from at least 1 Figure showing the markers and cellular functions associated with the markers. The same could be said for the cellular interactions described within the text. It was difficult to visualize the different cell types, roles in immunity, and cellular interactions using only the text provided.
We have added a figure to depict the four memory populations, their locations in the tissue and the cellular interactions with their environment. A Table has been added describing the function and expression of the memory T cell markers.
Specific comments
1) The manuscript would benefit from additional editing of the text, with emphasis on sentence structure.
The manuscript has been edited by an independent reader to improve the structure.
2) On line 84, the word “CD62Llow” should be corrected to “CD62low”
We believe the original wording is correct. The marker is CD62L. We have reformatted the text to make “low” in superscript.
